# One Bug, Hundreds Behind: LLMs for Large-Scale Bug Discovery

**Qiushi Wu** [1]   **Yue Xiao** [2]   **Dhilung Kirat** [1]   **Kevin Eykholt** [1]   **Jiyong Jang** [1]   **Douglas Lee Schales** [1]

## Abstract

Recurring Pattern Bugs (RPBs) are defined as bugs where a single root cause appears repeatedly across multiple code segments. These bugs remain a persistent security threat even after individual instances are patched. Various static analyzers exist for finding specific bug patterns but require significant engineering effort and fail to generalize well beyond their predefined template, preventing them from detecting RPBs. To tackle RPBs, we introduce BUGSTONE, a hybrid framework combining LLVM-based program analysis with Large Language Models to automate RPB detection. BUGSTONE leverages a single patched instance to synthesize abstract error patterns and retrieves semantically similar bugs throughout the codebase. To evaluate BUGSTONE, we create a ground truth dataset by analyzing over 1.9K security bugs reports, on which BUGSTONE achieves 92.2% precision and 79.1% pairwise accuracy. We further validated BUGSTONE through a large-scale real-world deployment. In the Linux kernel, BUGSTONE identified over 22K potential issues; a manual audit of 400 samples confirmed 246 valid bugs, including invalid pointer dereferences, resource leaks, type errors, performance issues, and others. To evaluate the generalizability of BUGSTONE, we further applied it to the top 100 Python projects, discovering multiple critical command injection vulnerabilities.

## 1. Introduction

Modern software ecosystems evolve at a rapid pace. For instance, the Linux kernel receives over 80K commits annually (Bai & Wu, 2023), growing to 28 million lines of code (AlDanial, 2024). However, such massive scale exposes a fundamental flaw in collaborative workflows. Under these conditions, vulnerabilities frequently recur as Recurring Pattern Bugs (RPBs), which are structurally similar flaws that persist throughout the codebase despite patches to the original instance. Consider a scenario where a specific API requires a resource release in an error handling path. While a developer may fix one instance of this missing release, identical logic often exists elsewhere in the program, obscured by different variable names or control flow structures. Because developers typically restrict their scope to the reported case, these latent variants remain unresolved, widening the attack surface and leading to duplicated maintenance efforts (He et al., 2023; Lin et al., 2023).

**The Gap: Over-specification vs. Lack of Grounding.** Detecting RPBs at scale remains an open challenge because it requires balancing structural precision with semantic generalization. Traditional static analysis tools (e.g., CodeQL, Coccinelle) offer high precision but suffer from over-specification. They require dedicated, customized code analyzers for each specific bug pattern under review. Moreover, these tools fail to generalize: a rule written to catch a missing lock issue in a `while` loop, for example, often fails to detect the same missing lock in a `for` loop or a `goto` statement, necessitating endless rule refinements (Zhou et al., 2022; Nielebock et al., 2020; Yun et al., 2016). In contrast, recent studies, such as Ullah et al. (Ullah et al., 2024), demonstrate that pure LLM-based methods still fall short in accurately detecting and analyzing bugs.

**Our Approach:** To bridge this gap, we propose BUG-STONE, a patch-centralized system that leverages a single patch as a clue to uncover all recurring pattern bugs within a codebase. BUGSTONE employs a five-stage pipeline: First, it normalizes the bug-fix commit with deterministic git-based preprocessing, removing non-semantic metadata and expanding the diff to the enclosing patched function. Second, BUGSTONE leverages a large language model (LLM) to summarize the details of the patch into a precise coding rule while pinpointing the API functions or code pieces responsible for the bug. Third, an LLVM-based static analyzer scans the compiled target program to enumerate call sites of the anchored API and maps them back to source-level caller functions. Fourth, BUGSTONE prompts the LLM to verify each retrieved candidate against the synthesized rule, optionally using the preprocessed seed patch as additional context. Finally, BUGSTONE aggregates and prioritizes po-

---

[1]IBM Research [2]William & Mary. Correspondence to: Qiushi Wu <qiushi.wu@ibm.com>.

*Proceedings of the $43^{rd}$ International Conference on Machine Learning*, Seoul, South Korea. PMLR 306, 2026. Copyright 2026 by the author(s).

tential violations for manual review before reporting them for patching. As a result, BUGSTONE can autonomously adapt to bugs with recurring patterns without the need to develop specialized tools tailored to individual bug patterns.

We evaluated BUGSTONE on the Linux kernel and the top Python projects, achieving 92.2% precision and 79.1% pairwise accuracy on the ground-truth dataset. To summarize, we made the following contributions:

- **An LLM-assisted program analysis framework for bug identification:** We propose BUGSTONE, the system to integrate lightweight static analysis with LLM reasoning for large-scale RPB detection. It successfully generalizes from single patch examples to find unseen variants.

- **BugStone-Bench Dataset:** We introduce and open-source a manually curated benchmark comprising 850 security patches across 80 recurring patterns (Bug-Stone, 2025). This dataset provides a rigorous ground truth for evaluating LLMs in software security tasks.

- **Systematic Evaluation of LLMs:** We benchmark six SOTA LLMs under various prompt configurations, providing empirical insights into the factors (e.g., prompt design, context retrieval) that drive performance in security tasks.

- **Real-World Impact:** BUGSTONE identified more than 22K potential bugs in the Linux kernel and 47 command injection issues in the top Python projects. To date, 34 findings have been confirmed and fixed by vendors, with 246 validated by expert manual review.

## 2. Related Work

**Traditional & ML-Based Bug Detection.** Static analysis tools (Bai et al., 2019; Lin et al., 2023; Tan et al., 2021; Wu et al., 2021) rely on manually crafted rules to detect specific patterns (e.g., sleep-in-atomic). However, they exhibit limited generalizability and demand substantial engineering overhead to adapt across diverse codebases. While dynamic analysis techniques (Kim et al., 2020; Ma et al., 2022; Bai et al., 2025; Jeong et al., 2019) can pinpoint memory vulnerabilities with high fidelity by monitoring runtime behaviors, they are inherently constrained by incomplete path coverage and substantial execution overhead. Early ML approaches employed Graph Neural Networks (GNNs) (Zhou et al., 2019; Cheng et al., 2021; Wu et al., 2024) or unsupervised learning (Ahmadi et al., 2021) to automate feature extraction. However, these methods require large, labeled datasets and often struggle with interpretability. Unlike these rigid or data-hungry approaches, BUGSTONE treats bug detection as an adaptive reasoning task, utilizing LLMs to interpret diverse code semantics without retraining or manual rule creation. Recent neuro-symbolic systems further reduce manual query engineering. For example, MoCQ (Li et al., 2025) uses LLMs to generate and refine executable Joern or CodeQL queries, preserving the determinism and low per-candidate cost of static analysis once a pattern is encoded. BUGSTONE makes a different tradeoff: static analysis only enumerates candidate locations, while the final semantic judgment is made by an LLM using a natural-language rule, improving flexibility across recurring patterns at the cost of LLM verification during deployment.

**LLM for Vulnerability Detection.** Recent studies evaluating off-the-shelf LLMs (Ullah et al., 2024; Yin & Ni, 2024; Gonçalves et al., 2025) reveal significant limitations in reasoning and localization, particularly for complex logic bugs. To mitigate this, researchers have explored enhancements such as RAG (Du et al., 2024), graph-based encodings (Lu et al., 2024), and domain-specific fine-tuning (Shestov et al., 2024). Closest to our work are GPTAid (Liu et al., 2025) and LLift (Li et al., 2024). However, GPTAid is restricted to parameter-level API misuse, whereas BUGSTONE captures complex, multi-statement recurring logic. Similarly, while LLift enhance static analysis specifically for Use-Before-Initialization (UBI) bugs, BUGSTONE leverages natural language rules to detect a broad spectrum of vulnerability classes (e.g., UAF, resource leaks) directly, identifying thousands of new issues rather than just improving path sensitivity.

**Semantics-Aware Clone Detection.** Code clone detection has evolved from token-matching to semantic-aware retrieval (Ragkhitwetsagul & Krinke, 2019; Shan et al., 2023). Vulnerability discovery tools like VulDeePecker (Li et al., 2018) and MVP (Xiao et al., 2020) leverage these techniques to map "code gadgets" or AST slices. However, these methods predominantly target *syntactic* clones (Type 1–3) or structural equivalents. They struggle with Type 4 semantic clones where implementation syntax differs entirely despite sharing the same logic. BUGSTONE addresses this limitation by abstracting code into natural language "coding rules" rather than relying on surface-level token or graph similarity, enabling the detection of logic variants that traditional clone detectors miss.

## 3. Methodology

### 3.1. Problem Formulation

This section establishes key definitions and explores the background of Recurring Pattern Bugs (RPBs).

#### 3.1.1. TERMINOLOGY

**Code Piece:** A continuous segment of source code such as a single macro, several basic blocks, or an entire function.

| API Name | Bug Pattern | Security Impact | Commit ID | Module | Patch Time | Found by |
|---|---|---|---|---|---|---|
| `iio_device_register_sysfs_group` | Missing release | Memory leak | 95a0d596bbd0 | drivers/iio/industrialio-core.c | Dec 8 2023 | Static Analyzer (Liu et al., 2024) |
| | | | 86fdd15e10e4 | drivers/iio/industrialio-event.c | Nov 15 2022 | Fuzzer |
| | | | 604faf9a2ecd | drivers/iio/industrialio-buffer.c | Oct 13 2021 | Fuzzer |
| `mlx5e_destroy_flow_table` | Missing nullification | Double free | 884abe45a901 | drivers/net/..../en_accel/fs_tcp.c | Jun 28 2023 | Unknown |
| | | | e75efc6466ae | drivers/net/..../en/fs_tt_redirect.c | Nov 28 2023 | Static Analyzer(Liu et al., 2024) |
| | | | 7a6eb072a954 | drivers/net/..../core/en_fs.c | Dec 28 2020 | Static Analyzer(Liu et al., 2021b) |
| `create_singlethread_workqueue` | Missing NULL check | NULL dereference | 1fdeb8b9f29d | drivers/net/..../iwlegacy/3945-mac.c | Feb 8 2023 | Static Analyzer(Jiang et al., 2024) |
| | | | 41f563ab3c33 | drivers/parisc/led.c | Nov 17 2022 | Fuzzing |
| | | | ba86af3733ae | drivers/net/...lan966x_ethtool.c | Nov 14 2022 | Fuzzing |
| | | | a82268b30a8b | drivers/infiniband/hw/nes/nes_cm.c | Feb 17 2016 | Static Analyzer (Yun et al., 2016) |
| `strncpy` | Using unsafe API | Overflow | de9b58400a3c | drivers/staging/..../ioctl_linux.c | Jul 12 2018 | Compile Warning |
| | | | 81b9de43599c | drivers/media/media-device.c | Jan 8 2018 | Compile Warning |
| | | | b3f8ab4b7953 | fs/9p/vfs_inode.c | Jul 16 2013 | Unknown |

*Table 1.* Examples of Recurring Pattern Bugs in the Linux kernel across four APIs. Similar issues needed to be fixed multiple times across the kernel.

**Recurring Pattern:** A repeated usage pattern of an API or code piece.

**Recurring Pattern Bugs (RPBs):** Recurring and similar errors that arise from misusing the same API, code piece, or code pattern.

**Security Coding Rule:** A concise statement specifying correct usage of an API or code piece; violating this rule may introduce security vulnerabilities.

**Seed Patch:** A representative patch that fixes a single instance of an RPB. It can be used as a concrete example to identify RPBs or to generate the corresponding security coding rule.

### 3.1.2. PREVALENCE AND PERSISTENCE OF RPBS

```
commit a82268b30a8b4b920d0bad24472cbb000c8e734a
Author: Insu Yun <wuninsu@gmail.com>
Date:   Wed Feb 17 13:06:33 2016 -0500

    nes: handling failed allocation when creating workqueue

    Since create_singlethread_workqueue uses kzalloc internally,
    it can fail when the system is under memory pressure, so need
    to handle it.

--- a/drivers/infiniband/hw/nes/nes_cm.c
+++ b/drivers/infiniband/hw/nes/nes_cm.c
@@ -2856,12 +2856,22 @@ static struct nes_cm_core *nes_cm_alloc_core(void)

    nes_debug(NES_DBG_CM, "Enable QUEUE EVENTS\n");
    cm_core->event_wq = create_singlethread_workqueue("nesewq");
+   if (!cm_core->event_wq)
+       goto out_free_cmcore;
    cm_core->post_event = nes_cm_post_event;
    nes_debug(NES_DBG_CM, "Enable QUEUE DISCONNECTS\n");
    cm_core->disconn_wq = create_singlethread_workqueue("nesdwq");
+   if (!cm_core->disconn_wq)
+       goto out_free_wq;

    print_core(cm_core);
    return cm_core;
+
+out_free_wq:
+   destroy_workqueue(cm_core->event_wq);
+out_free_cmcore:
+   kfree(cm_core);
+   return NULL;
```

*Figure 1.* The `create_singlethread_workqueue` macro can return `NULL` on failure. This patch adds a simple check to `nes_cm.c`. It is invoked more than 197 times in other parts of the Linux kernel, but not every invocation checks the returned pointer value. This creates an RPB.

Our analysis of the Linux kernel reveals that RPBs are not isolated incidents but a systemic issue driven by the complexity of modern APIs. **Table 1** presents four representative APIs where we identified recurring patterns. These examples illustrate that RPBs result in severe vulnerabilities, including memory leaks, NULL pointer dereferences (NPD), and Use-After-Free (UAF), requiring a series of patches over several years to address each specific issue.

**Origins of Recurrence:** RPBs typically stem from error-prone APIs that require specific, non-obvious safeguards. For instance, the macro `create_singlethread_workqueue` returns `NULL` on failure (see patch a82268b30a8b in Figure 1). Although straightforward, this macro is invoked 197 times in the kernel. This extensive usage frequently leads developers to omit the required null check, resulting in recurring NPDs across various modules.

**The Patch Gap:** A critical observation motivating our work is that *fixing one instance does not fix the pattern*. Due to modular maintenance, a developer often fixes the specific crash they encountered but leaves identical vulnerabilities in other drivers or subsystems untouched. As detailed in **Table 1**, bugs often persist for years after the initial fix because existing tools (fuzzers and static analyzers) lack the semantic understanding to connect the patch to other latent instances. This creates a persistent attack surface where known exploit patterns remain active in the codebase simply because they reside in a different file.

### 3.2. Workflow of BUGSTONE

Figure 2 illustrates the end-to-end workflow of BUGSTONE. First, a bug-fix commit is cleaned and expanded into a seed patch containing the full function context of the changed code. This step is deterministic and does not require compiling the patch. Second, the seed patch is processed by an LLM to synthesize a concise security coding rule (see Table 9), using manually written rule templates only as prompt guidance. Third, given the API or code element named in the rule, a lightweight LLVM-based analyzer scans the compiled target program to enumerate candidate call sites and extracts the corresponding caller functions as source-level

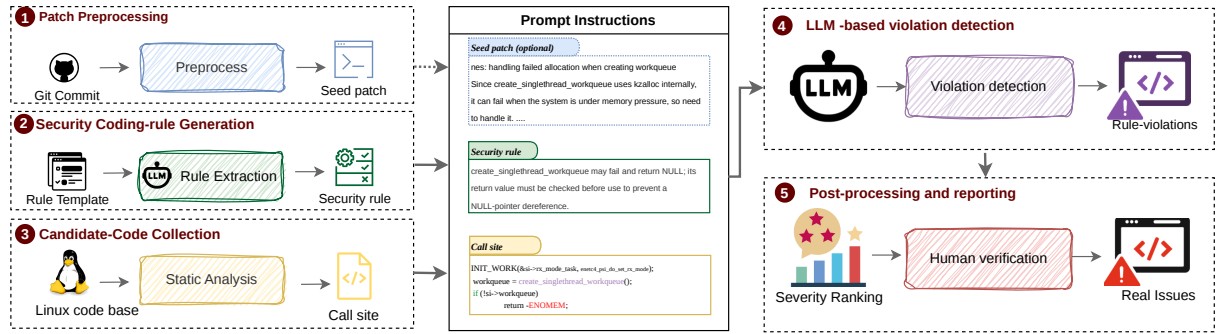

*Figure 2.* Overview of BUGSTONE.

context. Fourth, BUGSTONE prompts the LLM to verify each candidate against the generated security coding rule. Finally, potential violations are aggregated for manual review, ensuring that only verified bugs are reported for patching. Still using Figure 1 as an example, BUGSTONE processed the fix for using the API `create_singlethread_workqueue` and synthesized a security rule mandating NULL checks on its return value. A static scan retrieved 197 unverified call sites, which the LLM subsequently filtered down to 10 high-confidence candidates. Manual review identified four true violations, three of which are bugs similar to the one fixed by the seed patch.

### 3.3. The Neuro-Symbolic Analysis Pipeline

To overcome the discussed limitations of existing approaches in RPB detection, BUGSTONE is designed to meet three key requirements: (1) providing a systematic method to construct prompts and contexts that leverage the strengths of LLMs when analyzing code and detecting diverse types of RPBs, while minimizing manual and engineering effort, (2) ensuring the solution is both cost-efficient and effective, and (3) achieving accuracy that is comparable to state-of-the-art bug finding tools.

To meet these requirements, BUGSTONE implements a rule-centric, patch-seeded pipeline. Unlike rigid static analyzers, BUGSTONE uses LLMs for semantic abstraction, augmented by lightweight static analysis for efficient scope enumeration. The pipeline consists of five stages:

**Seed Patch Normalization.** Raw git commits contain both irrelevant metadata and insufficient local context. BUGSTONE therefore normalizes each seed patch with a deterministic preprocessing tool rather than an LLM or LLVM analysis. The tool first invokes git diff with function context (e.g., `git diff -W`) to recover the enclosing function body around each changed hunk. It then removes metadata that is not part of the code semantics, including commit hashes, author information, file headers, hunk markers, and commit tags. The normalized seed patch retains the com-

mit message, the relevant code changes, and the expanded function-level context. This gives the LLM enough local control-flow and data-dependency information to infer the intended fix, while avoiding manual normalization for the 850 seed patches in BugStone-Bench.

**Security Rule Synthesis.** To scale RPB detection, we abstract the specific fix into a generalized Security Coding Rule ($R$). While directly providing the seed patch ($P_{seed}$) to the LLM yields competitive accuracy, it incurs prohibitive token costs when scaling to large scale bug detection (see Section 6). Moreover, abstracting the patch into a concise natural language rule improves detection performance by decoupling the semantic logic of the vulnerability from the syntactic artifacts of the specific fix. We prompt the LLM to synthesize a rule that explicitly captures: (1) the Root Cause (the misused API), (2) the Required Action (e.g., "check return value"), and (3) Scope Constraints (e.g., "in error paths only"). The prompt contains three manually written rule templates as few-shot formatting guidance (Table 9). These templates are not manually instantiated for each patch; they only constrain the expected rule format. If none of the templates matches the patch, the LLM is instructed to emit a concise custom rule grounded only in the patch evidence. This abstraction allows BUGSTONE to maintain high precision while significantly reducing inference costs compared to raw patch processing.

**Candidate Collection with LLVM-Based Static Analysis.** After a rule identifies a target API or code element, BUGSTONE uses static analysis only to enumerate candidate code locations and recover their source-level context. The analyzer takes the entire target program as input and operates on LLVM IR produced by the program's normal build process. This compilation step naturally resolves preprocessor macros, configuration-dependent code, and in-line functions before analysis. We adapt MLTA (Lu & Hu, 2019), an LLVM-based call-graph analyzer, to construct a whole-program call graph and record the source-level file and line range of each function from LLVM debug meta-

data. Given a target sink function from the synthesized rule, BUGSTONE locates all call sites in the call graph, maps each call site to its enclosing caller function, and extracts that caller function's source code as the candidate snippet for LLM verification. Thus, the static analyzer prunes the search space and provides function-level context; it does not decide whether a candidate violates the rule. For non-C/C++ projects, such as Python, candidate enumeration is implemented with language-level source scanning for the specified sink APIs, while the same rule-based LLM verification stage is reused.

**Violation Detection Strategy.** Once the rule $R$ is synthesized and candidate functions $C = \{c_1, c_2, ...\}$ are retrieved via static analysis, BUGSTONE performs violation detection. We evaluate three strategies: (1) Patch-Based: instructing the LLM to determine if the candidate code exhibits the same vulnerability pattern present in the seed patch; (2) Rule-Based: assessing the candidate against the violation criteria of the security rule; (3) Hybrid: employing a one-shot prompting strategy where the seed patch serves as a concrete in-context demonstration, while the model evaluates the candidate against the specific constraints of the security rule. Empirical results in Section 5.1 demonstrate that the Rule-Based approach yields the best trade-off between accuracy and cost. By decoupling the "example" from the "logic," the LLM focuses strictly on semantic compliance rather than pattern matching.

**Impact-Based Prioritization.** Given the scale of findings (e.g., 22k findings in Linux), manual review of the entire result set is infeasible. BUGSTONE implements a severity ranking module to prioritize findings based on two factors: (1) Module Criticality: We classify kernel subsystems into four tiers. For example, core modules like `mm` (memory management) and `net` are prioritized over legacy architecture drivers. (2) Bug Class Severity: Vulnerabilities with higher exploitability (e.g., Null Pointer Dereference) are ranked above lower-impact issues (e.g., minor Memory Leaks). This filtering ensures that maintainers focus their effort on high-risk, high-impact vulnerabilities.

## 4. The BugStone-Bench & Evaluation Setup

To rigorously evaluate BUGSTONE, we constructed **BugStone-Bench**, a ground-truth dataset of Recurring Pattern Bugs derived from real-world security patches. We detail the dataset construction and the experimental protocols below.

### 4.1. Datasets Construction

#### 4.1.1. GROUND TRUTH DATASET CONSTRUCTION

We surveyed 23 security studies published at top-tier conferences that analyze the Linux kernel and report vulnerabilities (Lu et al., 2019; Wu et al., 2021; Pakki & Lu, 2020; Emamdoost et al., 2021; Liu et al., 2021b; Zhou et al., 2022; Liu et al., 2024; Wang et al., 2018; Wang, 2021; Yun et al., 2016; Min et al., 2015; Jeong et al., 2019; Zhai et al., 2022; Lyu et al., 2022; Tan et al., 2021; Suzuki et al., 2024; Dossche & Coppens, 2024; Lin et al., 2023; 2025; DeFreez et al., 2018; Kim et al., 2020; Xu et al., 2018; Bai et al., 2019). By matching authors' emails with kernel commit logs, we harvested **1,910** security patches contributed by 29 distinct authors. To ensure data quality, we engaged six security researchers, each holding a Ph.D. and possessing 6–30 years of domain experience, to manually inspect every patch. The experts filtered the dataset using the following taxonomy:

- **Recurring Pattern Bugs (RPB):** We identified 850 patches that form **80 distinct recurring patterns** (i.e., at least two patches address the same API or logic misuse).
- **Exclusions:** We excluded 39 independent patches (singletons) and 1,021 patches involving complex concurrency or multi-file dependencies outside the scope of function-level context.

**Test Set Formation:** For each of the 80 patterns, we randomly designated one patch as the *Seed Patch* (used for one-shot prompting or rule generation) and reserved the remaining patches for bug detection evaluation. For every reserved patch, we extracted both the *pre-fix* (vulnerable) and *post-fix* (safe) versions. This yielded a balanced test set of **1,540** code snippets ($2 \times (850 \text{ total} - 80 \text{ seeds})$).

#### 4.1.2. EXTENDED RULE SET CONSTRUCTION

To enable large-scale deployment beyond the held-out benchmark pairs, we compiled an extended set of 135 security coding rules through LLM-assisted rule synthesis followed by manual deduplication and validation. These rules are used in the real-world deployment in Section 6. They should not be confused with the HURULE baseline in Table 4. HURULE denotes human-written rules used only to compare LLM-generated rules with human-authored specifications in the benchmark evaluation. In the default BUGSTONE setting, including the RULE and RULE+PATCH rows in Table 4, the rule provided to the detector is generated from the seed patch by the LLM.

Specifically, these 135 security coding rules were derived from two sources: (1) Literature Review: We extracted 119 rules from the 1,910 expert-reviewed patches in our ground truth dataset. (2) Historical Mining: To identify novel recurring patterns, we also analyzed 1.06 million historical

| TRId | # Rules | Rule Template |
|------|---------|---------------|
| 1 | 54 | The function {TARGET} may fail and return {RETURN_VALUE}. Thus its return value should be checked before use to prevent {IMPACT} . |
| 2 | 57 | Once {TARGET} succeeds, ensure that {HANDLER} is invoked in the subsequent error-handling path to prevent {IMPACT}. |
| 3 | 9 | Use {HANDLER} instead of {TARGET1}+{TARGET2} to {GOAL}. |
| 4 | 2 | Memory allocated with {TARGET} must be freed with {HANDLER}, not kfree(). |
| 5 | 3 | After calling {TARGET}, the refcount is incremented regardless of success or failure, so {HANDLER} must be invoked in every error-handling path to prevent {IMPACT}. |
| 6 | 2 | Use {HANDLER} instead of {TARGET} when {CONDITION}, to prevent buffer overflow. |
| 7 | 1 | Use {HANDLER} instead of {TARGET} for Ethernet-address comparisons. This guarantees correct results and skips unnecessary bytewise memory checks. |
| 8 | 1 | Use {HANDLER} instead of {TARGET} for delays under 20ms |
| 9 | 1 | Release the {HANDLER} before calling {TARGET} and reacquire it immediately afterward to prevent {IMPACT}. |
| 10 | 1 | No need to call {TARGET} before destroying them with {HANDLER}, as it automatically drains them, thus avoiding unnecessary overhead. |
| 11 | 1 | The {TARGET} function returns an 'unsigned long' value instead of 'int'. Make sure the return value is put into a variable with unsigned long type. |
| 12 | 1 | {TARGET} returns a negative value on failure, so the return check should be irq ¡ 0 instead of irq == 0. |
| 13 | 1 | Instead of invoking {TARGET1} and {TARGET2} separately, use the {HANDLER} helper for iomap operations. This ensures proper resource management and avoids potential issues. |
| 14 | 1 | The third parameter passed to core_link_read_dpcd() may remain uninitialized if the call fails, and since that variable might later be used by functions like core_link_write_dpcd(), it should be zero (e.g., with memset) before invoking core_link_read_dpcd() to prevent undefined behavior. |

*Table 2.* 135 unique rules classified by template rule Id (TRId); # Rules: number of rules

Linux patches. Specifically, we leveraged Llama-3.3-8b to synthesize security rule summaries for each patch (see Table 9), followed by embedding-based clustering (using gpt4all (PyPI, 2025)) to group semantically similar fixes. We manually distilled the top 200 clusters representing approximately 17K patches into 34 representative rules. After merging overlaps, the final set comprises 135 unique rules distributed across 14 templates (see Table 2), covering diverse impacts from memory leaks to logic errors.

### 4.2. Experimental Setup

**Model Selection.** In this evaluation, we tested six LLMs with BUGSTONE using identical hyperparameters. Specifically, we set temperature, top-$p$, and $n$ to 1, following the default configuration of the OpenAI chat completion API (OpenAI, 2025a; Meta, 2025; Google, 2025). For state-of-the-art open-source models, we used models from the Llama family. For state-of-the-art closed-source models, we used models from the GPT and Claude families. Specifically, we evaluated Llama-3.3-8b and gpt-4.1-nano as lightweight, low-cost models; Llama-3.3-70b and Llama-4-17B-128E as mid-sized models that balance cost and accuracy; and o4-mini and Claude Sonnet 3.7 as higher-cost frontier models.

| Setup | Input | | | Strategy |
|-------|-------|-------|-------------|----------|
| | Code Snippet | Patch | Coding Rule | CoT |
| **Basic** | ✓ | | | ✓ |
| **Patch** | ✓ | ✓ | | ✓ |
| **HuRule** | ✓ | | Human | ✓ |
| **Rule** | ✓ | | System | ✓ |
| **Rule w/o CoT** | ✓ | | System | |
| **Rule + Patch** | ✓ | ✓ | System | ✓ |

*Table 3.* Prompt Configurations

**Prompt Configurations.** To isolate the contribution of each BUGSTONE component (see Table 3 and Table 10), we compared six strategies: (1) *Basic:* Zero-shot detection using only the code snippet; (2) *Patch:* Uses the seed patch

as a one-shot example; (3) *HuRule:* Uses a human-expert written rule, and note that, this is used only as an ablation baseline and is not used by the default system; (4) *Rule (Ours):* Uses the BUGSTONE-synthesized rule; (5) *Rule w/o CoT:* Same as (4) but without Chain-of-Thought reasoning; (6) *Rule+Patch (Hybrid):* Uses both the seed patch and synthesized rule.

**Evaluation Metrics.** For each prompt configuration and model, we report the precision, recall, and pairwise accuracy to quantify performance. Precision ($\mathcal{P}$) is the fraction of reported positives (i.e., flagged as an RPB) that are correct. Recall ($\mathcal{R}$) is the fraction of true bugs that are detected. Higher recall reflects better coverage and robustness. Pairwise accuracy ($\mathcal{PA}$) is a strict measure computed per patched and unpatched pair: the model earns credit only if it labels the patched snippet as safe and the unpatched snippet as buggy. Any other outcome counts the pair as incorrect. Thus, a random guess yields $\mathcal{PA} = 25\%$.

## 5. Evaluation

### 5.1. Performance and Ablation Study

Table 4 summarizes the performance of the six LLMs on our ground truth dataset under the six prompt configurations (see Table 10 for full prompt details). Each setup defines a distinct prompting scenario for the LLM to detect bugs in a code snippet.

**Basic Setup: High Failure Rate without Context.** Table 4 shows that, with the Basic configuration, most models achieved performance close to the random-guessing baseline of 25% $\mathcal{PA}$. Specifically, GPT-4.1-nano, Llama-4, Claude Sonnet 3.7, and o4-mini tend to label code as buggy, boosting recall but also generating more false positives, thereby causing humans to waste effort on verification during real-world bug finding. These results confirm prior findings that, without targeted guidance, LLMs struggle to accurately de-

| Setup | gpt-4.1-nano | | | Llama-3.3-8b | | | Llama-3.3-70b | | | Llama-4-17b-128E | | | Claude Sonnet 3.7 | | | o4-mini | | |
|---|---|---|---|---|---|---|---|---|---|---|---|---|---|---|---|---|---|---|
| Metrics | $\mathcal{P}$ | $\mathcal{R}$ | $\mathcal{PA}$ | $\mathcal{P}$ | $\mathcal{R}$ | $\mathcal{PA}$ | $\mathcal{P}$ | $\mathcal{R}$ | $\mathcal{PA}$ | $\mathcal{P}$ | $\mathcal{R}$ | $\mathcal{PA}$ | $\mathcal{P}$ | $\mathcal{R}$ | $\mathcal{PA}$ | $\mathcal{P}$ | $\mathcal{R}$ | $\mathcal{PA}$ |
| **Basic** | 54.3% | 78.1% | 20.0% | 49.4% | 30.5% | 16.8% | 53.8% | 57.7% | 22.0% | 53.5% | 83.4% | 17.3% | 58.2% | 81.8% | 29.1% | 61.3% | 85.1% | 37.3% |
| **Patch** | 69.5% | 72.2% | 50.9% | 71.4% | 55.1% | 39.0% | 90.7% | 67.4% | 62.6% | 88.0% | 72.5% | 63.9% | **91.2%** | 79.1% | 71.9% | 89.6% | 87.4% | 77.8% |
| **HuRule** | 80.4% | **84.5%** | 67.7% | **83.4%** | 67.8% | **57.9%** | 87.2% | **78.8%** | 69.4% | 88.8% | 81.4% | 72.8% | 88.5% | 87.7% | 76.7% | 88.7% | 90.9% | 79.9% |
| **Rule** | **82.2%** | **85.3%** | **70.1%** | 79.1% | 64.9% | 52.5% | 90.6% | 78.3% | 71.4% | **92.2%** | **85.8%** | 79.1% | 91.5% | 89.0% | 80.8% | 90.3% | 92.9% | 83.1% |
| **Rule w/o CoT** | 55.7% | 52.2% | 23.7% | 73.6% | 19.9% | 14.4% | **94.9%** | 7.3% | 7.3% | **92.9%** | **85.3%** | **79.5%** | 88.1% | 86.0% | 74.6% | 90.5% | **93.1%** | **83.5%** |
| **Rule+Patch** | 80.8% | 68.4% | 57.5% | 80.9% | 64.8% | 55.0% | 90.4% | 78.7% | 71.3% | 91.8% | 83.1% | 77.0% | 91.4% | **88.8%** | **80.9%** | **91.3%** | 92.7% | **84.1%** |

*Table 4.* Accuracy of LLMs under different setups ($\mathcal{P}$=Precision, $\mathcal{R}$=Recall, $\mathcal{PA}$=Pairwise Accuracy). Best results are bolded. Differences $< 1\%$ are also bolded and treated as similar performance due to non-determinism.

tect real-world bugs (Ullah et al., 2024). By contrast, the other five configurations, providing a patch, a human-crafted rule, an auto-formatted rule, or both patch and rule, substantially improve performance by enabling the model to focus on the relevant bug pattern.

**Patch Setup: High Cost, Moderate Accuracy.** The Patch configuration boosts $\mathcal{PA}$ by 22–47% over the baseline, confirming that one-shot examples aid RPB detection. However, this gain is model-dependent: while frontier models (e.g., Claude 3.7) leverage the extra context effectively, smaller models (e.g., Llama-8b) struggle to generalize. Crucially, this approach is inefficient compared to the Rule configuration. Also, including the raw patch increases the average input from approximately 0.87K to 2.25K tokens per query (see Table 6), while yielding lower precision due to patch-specific artifacts.

**Rule Setup: High Precision with Low Cost.** The Rule configuration yields the best $\mathcal{PA}$ and a high precision across most LLMs. Notably, Llama-4 achieves 92.2% precision and 79.1% $\mathcal{PA}$ under this configuration; given its strong accuracy, throughput, and cost profile, we select Llama-4 as our preferred model for large-scale RPBs detection (see Section 6). Therefore, for this project, we select the formatted rule configuration as our primary choice, because it requires lower cost, delivers higher accuracy across models, and supports easy automatic generation.

**Hybrid & HuRule Setups: Marginal Gains at High Cost.** Evaluations of alternative configurations reveal diminishing returns. The Hybrid setup (Rule+Patch) yields only modest improvements for frontier models (e.g., o4-mini) by reinforcing abstract rules with concrete examples; however, this comes at the expense of doubling token consumption, significantly degrading the cost-benefit ratio. The HuRule setup (Human-generated) aids smaller models (e.g., Llama-3.3-8b) due to its conversational phrasing, yet their absolute performance remains insufficient for reliable deployment compared to larger models using synthesized rules.

**CoT Ablation: Essential for Small Models, Redundant for Reasoning Models.** Removing Chain-of-Thought (CoT) instructions causes a performance collapse in smaller models (e.g., Llama-3.3-8b, gpt-4.1-nano), where pairwise accuracy drops to near-random levels ($\approx 25\%$). Without

explicit guidance to "analyze line-by-line," these models default to ungrounded guessing or stubborn "No" predictions. Conversely, reasoning-optimized models (e.g., o4-mini) remain resilient, maintaining high accuracy even without explicit CoT prompts. This is attributable to their architectural design, which performs implicit "hidden" reasoning prior to response generation, effectively rendering explicit CoT instructions redundant while reducing output token costs.

**Performance differences across LLMs.** In zero-shot settings, most models bias heavily towards false positives (answering "Yes"), with only o4-mini exceeding the random baseline. Even when provided with Rules or Patches, smaller models (e.g., Llama-3.3-8b, gpt-4.1-nano) fail to achieve reliable accuracy, indicating insufficient capacity for complex code reasoning. In contrast, larger models demonstrate superior efficacy with the Rule configuration. Notably, switching to the Patch configuration degrades performance across the board because models may attend to patch-specific artifacts rather than the abstract rule. However, reasoning-optimized models like o4-mini exhibit the highest resilience to this distraction, suffering the smallest accuracy drop.

### 5.2. Performance Stratification by Rule Template

| TRId | # of rules | # of cases | $\mathcal{P}$ | $\mathcal{R}$ | $\mathcal{PA}$ |
|---|---|---|---|---|---|
| 1 | 34 | 194 | 97.1% | 88.1% | 85.4% |
| 2 | 41 | 440 | 91.0% | 83.0% | 75.7% |
| 3 | 1 | 17 | 100% | 100% | 100% |
| 4 | 1 | 2 | 100% | 50% | 50% |
| 5 | 3 | 117 | 87.7% | 91.5% | 79.1% |
| **Overall** | **80** | **770** | **92.2%** | **85.8%** | **79.1%** |

*Table 5.* Accuracy of BUGSTONE for different types per template rule Id (TRId).

We also stratified the ground-truth performance by rule template based on Llama-4 with the rule configuration (see Table 5). BUGSTONE achieves near-perfect precision on **Templates 1** and **3**. These rules are easy to validate since they require confirming if the safeguard check or required API call was made. Performance drops slightly on **Templates 2** and **5**, which require deeper semantic reasoning over complex error-handling logic and control flow.

To understand this variance, we manually audited the failure

cases:

- **False positives** primarily arise from missing semantic context outside the extracted function. For example, a candidate may appear to violate a missing-check rule locally, but external API semantics or global state can make the code safe. In other cases, complex `goto`-based error paths cause the model to misread whether a check or handler dominates the relevant use.

- **False negatives** are driven by three factors: (1) inter-procedural context requirements, where accurate classification depends on callers, callees, or global state beyond the provided function; (2) dataflow complexity, where the model struggles to resolve coupled conditions such as conditional allocation requiring conditional release; and (3) long-function attention limits, where the model fails to enumerate all relevant paths.

### 5.3. Overhead and Cost

| | | Basic | Patch | HuRule | Rule | Rule w/o CoT | Rule + Patch |
|---|---|---|---|---|---|---|---|
| **Input Tokens** | | 0.84 | 2.25 | 0.87 | 0.87 | 0.86 | 2.29 |
| **Output Tokens** | gpt-4.1-nano | 1.27 | 0.93 | 0.75 | 0.80 | 0.004 | 0.86 |
| | Llama-3.3-8b | 0.94 | 0.44 | 0.48 | 0.52 | 0.05 | 0.46 |
| | Llama-3.3-70b | 0.73 | 0.48 | 0.49 | 0.51 | 0.11 | 0.48 |
| | Llama-4-17b-128E | 0.87 | 0.63 | 0.65 | 0.65 | 0.55 | 0.65 |
| | Claude Sonnet 3.7 | 0.79 | 0.64 | 0.52 | 0.54 | 0.20 | 0.59 |
| | o4-mini | 0.50 | 0.29 | 0.27 | 0.28 | 0.001 | 0.27 |

*Table 6.* Average input and output tokens per prompt configuration (in thousands)

We evaluate inference cost based on average token counts and standard vendor pricing (AI, 2025; OpenAI, 2025c; Anthropic, 2025).

**Input Costs.** As shown in Table 6, input size jumps from ∼850 tokens (Basic/Rule) to ∼2,250 tokens when including a seed patch, a 160% increase. This confirms that while one-shot prompting (Patch setup) provides context, it incurs a substantial cost penalty compared to the Rule-based approach.

**Output Costs.** Paradoxically, the *Basic* configuration yields the longest output tokens despite near-random accuracy, as models generate verbose, ungrounded rationales without context. And for a given model, average output length is often similar when the prompt allows for chain of thought (Patch, HuRule, Rule, Rule+Patch). From shortest to longest average output length the order is o4-mini, Llama-3.3, Claude Sonnet 3.7, Llama-4, and gpt-4.1-nano. In Figure 3, we show the average cost per patch (top) when using the Rule configuration prompt. Even though o4-mini has the shortest average output length, it is drastically more expensive than all but Claude Sonnet 3.7 due to its high unit price (AI, 2025; OpenAI, 2025c; Anthropic, 2025). In

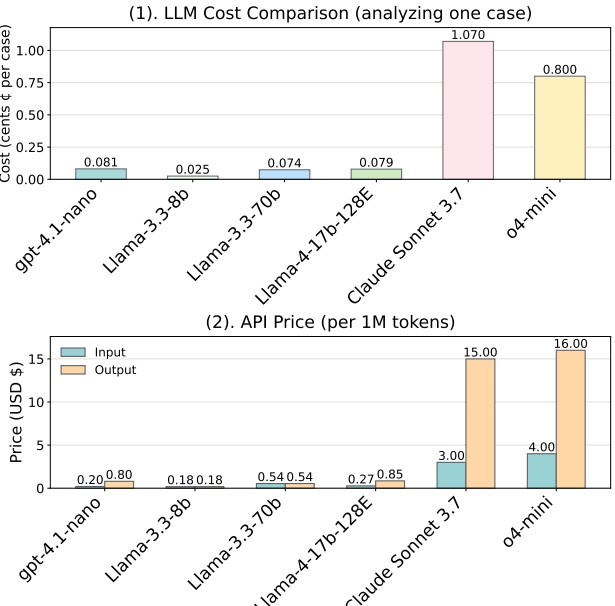

*Figure 3.* Average cost of analyzing one case with the Rule configuration prompt (top) and API call price of LLMs (bottom).

practice, costs are also likely to be lower when performing large scale bug detection as the prompt template and rules are fixed, allowing for input caching (OpenAI, 2025b). According to these results, Llama-4 is currently the ideal choice for RPB detection, given its competitive performance with frontier models, but at a much cheaper cost, under 0.1 cents per case. For the large-scale Linux deployment (see Section 6), BUGSTONE processed 148,664 candidate cases. Using Llama-4 at approximately $0.00074 per case, the estimated billed inference cost for the full Linux scan was about $110. During our experiments, we did not observe API retries, max-token failures, or malformed outputs requiring re-prompting during this scan.

## 6. Real-World Deployment

In this section, we use BUGSTONE to identify novel RPBs in the Linux kernel and top Python Projects.

### 6.1. Debugging the Linux Kernel

Using 135 synthesized rules, BUGSTONE scanned 148,664 code pieces (approximately 117M input tokens) and flagged 22,568 potential violations (Table 7). We randomly sampled 400 cases and, upon manual review, confirmed 246 as true violations (61.5% Precision), presenting diverse security implications. The most prevalent include:

- **Invalid Pointer Dereferences (22 cases):** e.g., missing NULL check for the return pointer of `kzalloc` (Figure 4).
- **Resource Leaks (80 cases):** e.g., unreleased objects or

| Target Project(s) | # violations discovered | # cases reviewed internally | # issues | # issues reported | # issues confirmed |
|---|---|---|---|---|---|
| Linux kernel | 22,568 | 400 | 246 | 40 | 32 |
| Python Projects | 47 | 47 | 20 | 5 | 2 |

*Table 7.* Real Bugs and Vulnerabilities Identified by BUGSTONE

memory in complex error paths.

- **Type & Logic Errors (85 cases):** e.g., numeric truncation in `wait_for_completion` (Figure 5) or performance degradation from `msleep` misuse (Figure 6).

As of submission, we have reported 40 issues to maintainers, with 32 confirmed and fixed. Although BUGSTONE exhibits a non-negligible false positive rate, it remains comparable to state-of-the-art static analyzers (Lin et al., 2023; Lu et al., 2019; Liu et al., 2021a; Wu et al., 2021) and manageable for manual triage.

**False Positive Analysis.** Our audit revealed two primary causes for false alarms: (1) **Contextual Correctness:** The code may technically violate a rule but remain safe due to external factors. For example, in `fs/ntfs3`, a missing check is safe because the `GFP_NOWARN` flag explicitly suppresses error reporting. Similarly, some resources (e.g., in `init_urbs`) are freed by caller functions, which BUGSTONE's intra-procedural view misses. (2) **Intentional Design:** Module initialization routines (e.g., `t_sdata_init`) often intentionally omit error handling, preferring to let the kernel panic on boot failure rather than complicate the code with recovery logic. During our case study, we did not observe many false positives caused by the model inventing rules unrelated to the prompt; most errors came from absent inter-procedural or project-specific semantic context.

### 6.2. Debugging Python Projects

To evaluate BUGSTONE's adaptability beyond C/C++, we targeted **Command Injection (CWE-78)** in the top 100 Python repositories (Evan & Kfir, 2025) (produced on Oct 20, 2025). We synthesized a security rule enforcing input validation for sink functions like `os.system` and `subprocess.run`, instructing the model to flag cases where user-controlled input reaches these APIs without sanitization.

**Results.** The second row of Table 7 summarizes our current progress. BUGSTONE flagged 47 violations in the collected Python projects. Of these cases, we have created proofs of vulnerability for 20 of them, which may allow for remote code execution depending on the deployment model. Figure 7 shows an example flagged by BUGSTONE that was confirmed and fixed by the product maintainers via private communication. The vulnerability occurred in an industrial product, PaddleOCR (Cui et al., 2025; PaddleTeam & Baidu, 2025), which provides online Optical Character Recognition

(OCR) services. It allowed an attacker to execute arbitrary commands on the server by providing a file with a malicious payload in its file name. Beyond this instance, our team is reviewing additional similar command injection vulnerabilities and will report them securely once they have been internally verified.

### 6.3. Limitations

BUGSTONE verifies each candidate with only the enclosing function, which keeps scans affordable but misses caller-side cleanup, callee-side ownership transfer, global flags, and module-level policies. It is also limited to recurring patterns anchored to an API, macro, helper, or code element named by the synthesized rule; unanchored arithmetic or concurrency bugs require different retrieval mechanisms. We pass the full caller function rather than a generic slice because control-flow context that appears unrelated can determine whether a check or cleanup is required. Since the final verifier is an LLM, the system has no soundness or completeness guarantee and all findings require manual review.

### 7. Conclusion

This paper introduces BUGSTONE, a neuro-symbolic framework that combines lightweight static analysis with LLM-guided reasoning to detect recurring pattern bugs (RPBs) from a single exemplar fix. By cleaning and expanding the context of a seed patch, BUGSTONE synthesizes standardized security coding rules that capture the root cause and required handling logic, enabling the detection of semantically similar bugs across massive codebases.

Our evaluation on **BugStone-Bench** (BugStone, 2025), a ground-truth dataset of 850 security patches, demonstrates that BUGSTONE achieves **92.2% precision** and **79.1% pairwise accuracy** using the rule-based configuration. Deployed at scale with 135 synthesized rules, BUGSTONE identified 246 manually validated issues in the Linux kernel (including invalid pointer dereferences and resource leaks) and 20 exploitable command injections in top Python projects. These results highlight BUGSTONE's ability to lower the barrier for RPB detection, allowing for the identification of complex path-sensitive bugs without the engineering cost of designing specialized static checkers.

## Impact Statement

As shown in Section 6.1 and Section 6.2, BUGSTONE revealed thousands of new issues in the Linux kernel and major Python projects, which are being responsibly disclosed to vendors. However, during our internal review, we observed that the severity of these findings varies greatly. Some are severe and exploitable, others less critical, and a few valid cases pose no immediate risk (e.g., initialization functions robust to failures (Linux, 2021)). We also notice some cases that had already been discussed but not merged, often due to maintainer prioritization, module-specific semantics, or policies that focus only on dynamically exploitable issues. Given the large and diverse set of findings, reporting all cases would be neither practical nor responsible and could overwhelm maintainers.

Therefore, we have established the following procedure to handle and report the findings generated by BUGSTONE. First, a post-processing module (see Section 3.3) ranks flagged violations by their severity and potential security impact. High-priority cases undergo detailed manual auditing by our research team. For Python projects, after verifying a finding, we additionally attempt to manually exploit command injection vulnerabilities so as to provide a proof of vulnerability (PoV) typically required for reporting issues in user-space programs. Validated Linux cases are disclosed to internal Linux maintainers within our organization for additional expert review and security risk assessment. After that, we proceed with responsible disclosure to the corresponding external Linux kernel maintainers. Python vulnerabilities are reported to vendors through private channels, such as GitHub's vulnerability reporting interface or direct emails to maintainers. As bug validation and responsible disclosure are time-consuming, this work is ongoing, and the numbers in Table 7 will be updated accordingly.

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

## A. Appendix

| TRId | # Rules | Rule Template |
|---|---|---|
| 1 | 54 | The function {TARGET} may fail and return {RETURN_VALUE}. Thus its return value should be checked before use to prevent {IMPACT} . |
| 2 | 57 | Once {TARGET} succeeds, ensure that {HANDLER} is invoked in the subsequent error-handling path to prevent {IMPACT}. |
| 3 | 9 | Use {HANDLER} instead of {TARGET1}+{TARGET2} to {GOAL}. |
| 4 | 2 | Memory allocated with {TARGET} must be freed with {HANDLER}, not kfree(). |
| 5 | 3 | After calling {TARGET}, the refcount is incremented regardless of success or failure, so {HANDLER} must be invoked in every error-handling path to prevent {IMPACT}. |
| 6 | 2 | Use {HANDLER} instead of {TARGET} when {CONDITION}, to prevent buffer overflow. |
| 7 | 1 | Use {HANDLER} instead of {TARGET} for Ethernet-address comparisons. This guarantees correct results and skips unnecessary bytewise memory checks. |
| 8 | 1 | Use {HANDLER} instead of {TARGET} for delays under 20ms |
| 9 | 1 | Release the {HANDLER} before calling {TARGET} and reacquire it immediately afterward to prevent {IMPACT}. |
| 10 | 1 | No need to call {TARGET} before destroying them with {HANDLER}, as it automatically drains them, thus avoiding unnecessary overhead. |
| 11 | 1 | The {TARGET} function returns an 'unsigned long' value instead of 'int'. Make sure the return value is put into a variable with unsigned long type. |
| 12 | 1 | {TARGET} returns a negative value on failure, so the return check should be irq ¡ 0 instead of irq == 0. |
| 13 | 1 | Instead of invoking {TARGET1} and {TARGET2 } separately, use the {HANDLER} helper for iomap operations. This ensures proper resource management and avoids potential issues. |
| 14 | 1 | The third parameter passed to core_link_read_dpcd() may remain uninitialized if the call fails, and since that variable might later be used by functions like core_link_write_dpcd(), it should be zero (e.g., with memset) before invoking core_link_read_dpcd() to prevent undefined behavior. |

*Table 8.* 135 unique rules classified by template rule Id (TRId); # Rules: number of rules

**Prompt for generating the security coding rule**

You are a software security expert. Your task is to generate one or more security coding rules from the given patch. Definition: A security coding rule is a concise statement that specifies the correct usage of a target API or code element. Violating this rule can introduce bugs or other issues.Use the templates below whenever they fit the patch. If none fit, write your own concise rule in a single sentence.

[Security Coding Rule Templates]

1. The function {TARGET} may fail and return {ERR_RETURN_VALUE}. Therefore,its return value must be checked before use to prevent {IMPACT}.

2. Once {TARGET} succeeds, ensure that {HANDLER} is invoked in any subsequent error handling path to prevent {IMPACT}.

3. Use {HANDLER} instead of {TARGET1} + {TARGET2} to {GOAL}.

[END Security Coding Rule Templates]

Output requirements:

1. Derive rules only from evidence in the patch, do not speculate.

2. Use identifiers as they appear in the patch.

3. If the patch addresses multiple independent issues, output multiple rules, one per line.

4. Do not include explanations or restate the patch.

[PATCH]

{PUT_PATCH_HERE}

[END PATCH]

Please provide the security coding rule or rules using the templates when possible. If no template fits, provide a concise custom rule.

*Table 9.* Prompt for generating security coding rule. The placeholder {PUT_PATCH_HERE} should be replaced with a patch.

| Prompt Setup | Prompt |
|---|---|
| **Basic** | You will be provided with a code. Your task is to analyze the code by following each code path and identify if the code contains any bugs, such as NULL pointer dereference, memory leak, refcount leak, etc.
[Code Snippets] {TARGET_CODE} [end]
Analyze the code line by line and show the analyzing steps. Finally, respond with 'YES' if there must be a bug in the given code, or 'NO' otherwise. Do not assume other situations that are not appeared in the code. |
| **Patch** | You will be provided with a patch. Afterwards, you will be shared with a code. Your task is to analyze the code by following each code path and identify if the code contains the same issue addressed by the patch.
[Patch] {PATCH} [end]
[Code Snippets] {TARGET_CODE} [end]
Analyze the code line by line and show the analyzing steps. Finally, respond with 'YES' if there must be a similar bug in the given code, or 'NO' otherwise. Do not assume other situations that are not appeared in the code. |
| **Rule or HuRule** | You will be provided with a security coding rule. Afterwards, you will be shared with a code. Your task is to analyze the code by following each code path and identify if the code violate the security rule.
[Security coding rule] {RULE} [end]
[Code Snippets] {TARGET_CODE} [end]
Analyze the code line by line and show the analyzing steps. Finally, respond with 'YES' if there must be a violation of the coding rule, or 'NO' otherwise. Do not assume other situations that are not appeared in the code or not mentioned in the coding rule. |
| **Rule w/o CoT** | You will be provided with a security coding rule. Afterwards, you will be shared with a code. Your task is to analyze the code by following each code path and identify if the code violate the security rule.
[Security coding rule] {RULE} [end]
[Code Snippets] {TARGET_CODE} [end]
Respond with 'YES' if there must be a violation of the coding rule, or 'NO' otherwise. Do not assume other situations that are not appeared in the code or not mentioned in the coding rule. |
| **Rule+Patch** | You will be provided with a security coding rule, its related patch. Afterwards, you will be shared with a code. Your task is to analyze the code by following each code path and identify if the code violate the security rule.
[Security coding rule] {RULE} [end]
[Patch] {PATCH} [end]
[Code Snippets] {TARGET_CODE} [end]
Analyze the code line by line and show the analyzing steps. Finally, respond with 'YES' if there must be a violation of the coding rule, or 'NO' otherwise. Do not assume other situations that are not appeared in the code or not mentioned in the coding rule. |

*Table 10.* Prompt Configurations. Replace the placeholders { } with a security coding rule, a seed patch, or a target code fragment for analysis.

```
1 link = kzalloc(sizeof(*link), GFP_KERNEL);
2 ...
3 ret = iwl_mld_init_link(mld, bss_conf, link);
4 ...
```

*Figure 4.* Real-World Case Example in the Linux kernel: Missing NULL check.

```
1 unsigned long __sched
2 wait_for_completion_timeout(struct completion *x,
3          unsigned long timeout)
4 {
5      return wait_for_common(x, timeout, TASK_UNINTERRUPTIBLE);
6 }
7
8 int rs_size, res;
9 ...
10 res = wait_for_completion_timeout(&ipmi->read_complete,
11     IPMI_TIMEOUT);
12 ...
```

*Figure 5.* Real-World Case Example in the Linux kernel: Numeric Truncation.

```
1 while (true) {
2     ...
3     msleep(5);
4 }
```

*Figure 6.* Real-World Case Example in the Linux kernel: Performance.

```
1 # PaddleOCR/ppstructure/pdf2word/pdf2word.py
2 selectedFiles = QFileDialog.getOpenFileNames(
3     self,
4     "Multiple file selection",
5     "/",
6     "Image file (*.png *.jpeg *.jpg *.bmp *.pdf)"
7 )[0]
8 if len(selectedFiles) > 0:
9     self.imagePaths = selectedFiles
10
11 ...
12
13 self.output_dir = os.path.join(
14     os.path.dirname(self.imagePaths[0]), "output"
15 )  # output_dir should be same as imagepath
16
17 ...
18
19 os.system("open " + os.path.normpath(self.output_dir))
```

*Figure 7.* Real-World Case Example in Python: Command Injection.

