# OpenReview forum: "One Bug, Hundreds Behind: LLMs for Large-Scale Bug Discovery"
_ICML.cc/2026/Conference — ICML 2026 regular_

### Official Review · Reviewer_PhbB · 2026-03-05

**Soundness:** 2
**Presentation:** 3
**Significance:** 2
**Originality:** 2
**Overall Recommendation:** 3
**Confidence:** 3

**Summary:**

The paper introduces BUGSTONE, a patch-centric framework that combines static analysis with LLM reasoning to automatically discover RPBs at scale. Based on a detected code patch, BUGSTONE expands contextual code using static analysis, derives a generalized security rule via an LLM, retrieves structurally similar code regions, and then uses the LLM to determine whether those instances share the same vulnerability. This design enables semantic generalization without requiring custom analyzers for each bug pattern.

The authors also introduce BugStone-Bench, a dataset comprising 850 security patches across 80 bug patterns. In an evaluation involving six different LLMs, the framework achieved 92.2% precision and 79.1% pairwise accuracy. To demonstrate real-world efficacy, the authors identified over 22,000 potential issues in the Linux kernel; a manual review of 400 samples confirmed 246 valid vulnerabilities. Furthermore, applying BUGSTONE to the top 100 Python projects revealed several critical command injection vulnerabilities.

**Compliance With Llm Reviewing Policy:**

Affirmed.

**Final Justification:**

I am not convinced by the answers to my follow-up questions, so I stay as a weak reject. For example, Q1 is a fundamental limitation of their approach. Of course, it is a limitation of intra-procedural analysis, but going intra-procedural is a design choice the authors made. The answer to Q2 doesn't justify the lack of baselines. If evaluation on the latest kernel is not viable,  the authors can evaluate on an older version or a smaller alternative artifact. I also found the answers to Q3 and Q4 unsatisfactory.

**Key Questions For Authors:**

1. How does BUGSTONE’s LLM-driven rule derivation perform when the provided code context contains significant "noise" or logic that is irrelevant to the RPB? Did you measure the degradation in precision when the input snippet size increases?

2. The authors noted that BUGSTONE’s performance is comparable to traditional static analysis but with a higher false-positive rate. In your manual review of 400 Linux kernel samples (where 246 were valid), what were the primary reasons causes for the 154 false positives? Were they caused by LLM "hallucinations" regarding the security rule, or by the static analyzer failing to provide sufficient control-flow context?

3. Could the authors provide the total API costs incurred when scanning the Linux kernel and the 100 Python projects?

4. Could the authors provide a comparative evaluation against other state-of-the-art bug detection tools (both LLM-based and non-LLM-based)? How does BUGSTONE perform in terms of Recall and Precision compared to these baselines?

5. What are the technical differences between BUGSTONE's method and the method in "Automated Static Vulnerability Detection via a Holistic Neuro-symbolic Approach"?

**Limitations:**

No, they should discuss the potential precision degradation when the code context increase,  and the API cost when applying their method to real-world applications.

**Strengths And Weaknesses:**

**Strengths**

- Scalability and Methodology: The work proposes a method for identifying RPBs on a large scale. While its performance profile differs from traditional tools (e.g., a higher false-positive rate), its effectiveness remains comparable to established static analysis tools.

- Practical Validation: The authors successfully identified novel RPBs within high-profile targets like the Linux kernel and major Python projects, demonstrating the framework's practical utility in real-world software ecosystems.

**Weaknesses**

- Experimental Design Flaws: There is a potential concern regarding the evaluation methodology. The experimental data relies on 80 identified rules and their corresponding code patches, calculating accuracy by applying LLMs to code snippets before and after a fix. In a production environment, it is rarely possible to isolate a code patch that perfectly aligns with a given rule via static analysis. Real-world code often contains significant "noise"—logic unrelated to the specific bug pattern. It remains unclear whether the method can maintain its performance when faced with such atmospheric complexity.

- Operational Overhead: While the authors provide the API cost for processing a single case, they do not specify the total volume of cases encountered during real-world application. This makes it difficult to estimate the actual financial or computational overhead in practice.

- Lack of Comparative Baselines: The evaluation lacks a direct comparison with existing bug-finding methodologies. Specifically, the paper does not provide performance metrics against traditional static analysis tools (e.g., CodeQL) or recent LLM-based vulnerability detection methods. Without these baselines, it is difficult to evaluate the actual improvement or competitive advantage of BUGSTONE.

- Ambiguous Novelty: The paper does not sufficiently compare itself with existing relative works, such as "Automated Static Vulnerability Detection via a Holistic Neuro-symbolic Approach". Since both works leverage LLMs and static analysis for vulnerability detection, the contribution and novelty of BUGSTONE compared to this work are not clearly discussed.

---

> ### Author Rebuttal · Authors · 2026-03-31
>
> We appreciate your careful reading and constructive feedback. We respond to each question and concern below.
> ### For Q1:
> First, we apologize for the confusion regarding rule extraction; further details are provided in Section 3.3 (Seed Patch Normalization). In brief, a deterministic tool is used for context expansion and cleaning of non-code metadata such as git tags and author information. The LLM, not a static analyzer, then extracts security coding rules describing the patch behavior based on the patch description, code changes, and surrounding context.
> For bug detection, we did not specifically measure the relationship between input snippet size and performance, as isolating this variable is difficult in practice. Even for the same bug type, longer contexts may stem from either bug-relevant or irrelevant code, making it hard to attribute performance changes solely to context size. BUGSTONE scans the entire target function without a dedicated noise reduction step, so the reported results already reflect real-world code complexity and noise levels.
> However, indeed, semantic complexity does influence performance. Table 4 shows that BUGSTONE performs lower on TRId 2 than TRId 1 (see Table 7), as localizing a handler along an error-handling path is more demanding than verifying a return value sanity check. Nevertheless, LLaMA-4 level models still achieve acceptable accuracy on these complex cases. Model capability also plays an important role, as smaller models and the absence of chain-of-thought reasoning noticeably degrade performance (see Table 3). As LLMs continue to advance, these limitations are expected to diminish accordingly.
>
> ### For Q2:
> The primary causes of false positives (FPs) are discussed in Section 6.1. The majority of FPs stem from insufficient context and incomplete semantic understanding, largely attributable to the limitations of intra-procedural analysis and the absence of global program state awareness.
> We did not observe significant LLM hallucinations in the FP cases. We attribute this to the use of concrete prompt instructions paired with the corresponding code context, as well as the enabling of chain-of-thought reasoning, which appears to ground the model's judgments in the provided evidence rather than spurious inferences.
>
> ### For Q3:
> Figure 3(1) reports the per-case cost of analyzing a single instance with different LLMs. For the large-scale bug detection described in Section 6.1, BUGSTONE processed a total of 148,664 cases in the Linux kernel. Using Llama-4 at approximately  0.00074 USD per case, the total cost for scanning the entire Linux kernel was roughly 110 USD. For the Python projects, the per-case cost is comparable with small variations due to differences in input context length. While we did not track the total aggregate cost, it can be estimated by multiplying the number of scanned cases per sink function by the per-case cost. For example, in total, 169 cases were scanned for calls to os.system and subprocess.run, corresponding to a total cost of approximately 0.13 USD. We can update the final version of the paper with these values as well.
>
> ### For Q4:
> Please refer to our response to Question 2 of Reviewer 2CD5, which addresses this concern in detail. We omit a duplicate response here due to rebuttal space constraints.
>
> ### For Q5:
> MoCQ uses an LLM to generate and iteratively refine queries for a static analyzer to identify bugs. BUGSTONE, by contrast, derives security coding rules from known patches and uses an LLM directly for bug detection, with static analysis serving only to reduce the candidate search space.
> The key advantage of BUGSTONE is flexibility. Detection logic is expressed in natural language rules, allowing BUGSTONE to be adapted to new bug patterns and target projects without reimplementing any analysis. MoCQ's generated query code is deterministic and incurs lower cost at detection time, but generating and refining it requires LLM invocations and must be repeated for each new target project. Both MoCQ and traditional static analyzers require non-trivial modification to generalize across projects.
>
> To illustrate: for detecting a missing return value check on an allocation function, BUGSTONE derives a natural language rule from existing patches, extracts all call sites via static analysis, and has the LLM evaluate each case. A traditional static analyzer or MoCQ must instead formally model sanity checks, distinguish error-handling from normal execution paths, and perform dataflow and alias analysis, all of which are project-dependent. For example, error-handling in the Linux kernel uses specific error codes such as ENOMEM, while OpenSSL or FreeBSD may follow entirely different conventions. Adapting such an approach to a new target requires substantial engineering effort, whereas BUGSTONE requires none.

---

> > ### Author Rebuttal · Reviewer_PhbB · 2026-04-02
> >
> > Thank you for your detailed rebuttal. While the explanations provide some clarity on the operational mechanics of BUGSTONE, several critical concerns regarding the evaluation methodology, comparative baselines, and claims regarding prior work remain unresolved. Please clarity on the following points:
> >
> > 1. The Relationship Between Noise, Context Length, and False Positives (Ref. Q1 & Q2)
> >
> > In response to Q1, you noted that BUGSTONE scans the entire target without a dedicated noise reduction step, and you hypothesize that future LLM advancements will mitigate the performance drop on complex semantic tasks. In response to Q2, you attribute the majority of False Positives (FPs) to "insufficient context and incomplete semantic understanding."These two points appear inherently linked. By passing noisy, unfiltered context to the LLM, the model's ability to maintain focus and reason over the relevant control-flow paths is predictably degraded. Relying on the assumption that future models will naturally solve this is not a substitute for a robust methodology today.
> >
> > Follow-up questions: Given that the lack of noise reduction directly exacerbates the "incomplete semantic understanding" noted in Q2, why is an intermediate pruning or slicing step (to isolate the core logic of the RPB) omitted? Can you provide any empirical evidence or ablation isolating how the ratio of irrelevant-to-relevant tokens in your 148,664 Linux kernel cases correlates with the FP rate?
> >
> > 2. Absence of Comparative Baselines (Ref. Q4)
> >
> > The refusal to provide a comparative evaluation against state-of-the-art tools is highly problematic. While I acknowledge your argument that BUGSTONE is designed to be a flexible, multi-pattern tool whereas traditional static analyzers are often designed for single-bug patterns, this does not preclude a fair comparison.
> >
> > Follow-up questions: To establish the efficacy of BUGSTONE, it must be benchmarked against existing tools on the intersecting set of bug patterns that both tools can detect. How does BUGSTONE's Precision and Recall compare to traditional static analysis on a specific, shared bug pattern? Without this baseline, it is impossible to quantify the trade-off between BUGSTONE's natural language flexibility and the potential loss of rigorous analytical precision.
> >
> > 3. Mischaracterization of MoCQ (Ref. Q5)
> >
> > In your rebuttal, you contrast BUGSTONE with MoCQ by stating that MoCQ's approach "must be repeated for each new target project" and that adapting it "requires substantial engineering effort." This directly contradicts the claims made in the MoCQ literature. The authors of MoCQ explicitly state that their approach is "language-agnostic," "applicable to a wide range of languages and vulnerability types," and that it induces generalized patterns from seed inputs "without requiring any changes to its design."
> >
> > Follow-up questions: Could you clarify exactly why MoCQ would require "substantial engineering effort" to generalize across projects, given that it relies on LLMs to synthesize general, extensible queries from a few vulnerability examples? Please provide a specific technical comparison of the generalizability limits of MoCQ's generated queries versus BUGSTONE's natural language rules.
> >
> > 4. Empirical vs. Estimated API Costs (Ref. Q3)
> >
> > While the estimated cost of 110 USD for the Linux kernel scan is relatively low, relying on a post-hoc mathematical estimation (average per-case cost $\times$ total cases) rather than empirical tracking obscures the actual operational overhead.
> > Follow-up questions: Does this estimation account for real-world execution anomalies, such as LLM API retries, max-token limit errors, or malformed outputs that required re-prompting during the large-scale scan? Please clarify if the 110 USD represents the theoretical minimum or the actual billed cost of the experiment.

---

> > > ### Author Response · Authors · 2026-04-06
> > >
> > > We thank for your thoughtful and detailed questions.
> > >
> > > ### For Q1, Insufficient context:
> > > We attribute major False Positives (FPs) to insufficient context but not noises. Insufficient context refers to information absent from the function body entirely, such as global variables. A concrete example: in the Linux kernel, fs/ntfs3/super.c, an unchecked return value of  kmemdump appears to violate the security coding rule, yet no bug exists because the call passes flag GFP\_NOWARN, which suppresses error handling by design. Without proving such flag semantics, the LLM may not make the correct determination. This is a fundamental limitation of intra-procedural analysis, not a consequence of noisy input, and addressing it requires interprocedural reasoning, which we identify as future work.
> > >
> > > ### For Q1': lack of noise reduction and irrelevant tokens:
> > > The noise reduction step of BUGSTONE applies only to the seed patch processing step for security coding rule generation, not to bug detection. Irrelevant tokens in our framework refer exclusively to non-code and bug related metadata in patch, such as "commit a82268b...", "--- a/drivers/...", "@@ -2856,12 @@" as shown in Figure 1. In the detection phase, the function body is passed as-is, intentionally reflecting real-world code complexity in the Linux kernel and the tested programs.
> > >
> > > We did experiment with program slicing and context windowing, though this is not discussed in the paper. These approaches did not perform well in practice, as determining what constitutes relevant context is itself non-trivial. For example, even a variable with no direct dataflow relationship to the sink API may remain semantically relevant due to control flow dependencies, such as influencing a branch condition that governs whether the API's return value requires checking. Defining a principled and generalizable slicing criterion that captures such relationships remains an open challenge.
> > >
> > > ### For Q2:
> > > As detailed in our response to Reviewer 2CD5, we evaluate BUGSTONE against existing approaches using the ground truth dataset in Section 4.1.1, comprising thousands of bugs from 23 top-tier security works relying on static or dynamic analyzers. The precision, recall, and accuracy results in Table 3 reflect BUGSTONE's performance on bugs identified by these tools, and Section 6.1 further demonstrates its ability to detect previously unreported bugs missed by all prior approaches.
> > >
> > > Re-running existing tools on the latest kernel is a significant engineering challenge, as analyzers depend on specific LLVM versions, and kernel updates may break compatibility. For example, LLVM's transition to opaque pointers forces static analyzers to abandon direct type querying, requiring significant implementation or even design changes when adapting tools built on older LLVM versions. This is also why state-of-the-art static analyzers rarely compare directly against one another: [1,2,4] include no such comparisons, [3] includes one comparison but acknowledges significant limitations (see their Appendix C), and [5] compares only theoretically.  We also acknowledge that BUGSTONE has varying performance across bug types, which is why we include Table 4 and Section 5.2 to characterize its scope and limitations. And also, the BUGSTONE does not cover interprocedural or concurrent use-after-free bugs as identified in [1].
> > >
> > > [1] Bai, Jia-Ju, et al. "Effective static analysis of concurrency Use-After-Free bugs in linux device drivers." USENIX ATC 19.
> > >
> > > [2]. DeFreez, Daniel, Aditya V. Thakur, and Cindy Rubio-González. "Path-based function embedding and its application to error-handling specification mining." Proceedings of the ESEC/FSE 2018.
> > >
> > > [3]. Dossche, Niels, and Bart Coppens. "Inference of error specifications and bug detection using structural similarities." USENIX Security 24
> > >
> > > [4]. Lu, Kangjie, Aditya Pakki, and Qiushi Wu. "Detecting Missing-Check bugs via semantic-and Context-Aware criticalness and constraints inferences." USENIX Security 19
> > >
> > > [5]. Wang, Wenwen. "MLEE: Effective Detection of Memory Leaks on Early-Exit Paths in OS Kernels." USENIX ATC 21
> > >
> > > ### For Q3:
> > > MoCQ and BUGSTONE are fundamentally different: MoCQ automates the generation of static analyzer queries, whereas BUGSTONE performs LLM-driven code auditing directly. The "substantial engineering effort" we referred to was directed at traditional static analyzers in general for bug findings and handling project-specific semantics. We will provide a more precise comparison in the final version upon acceptance.
> > >
> > > ### For Q4 :
> > > Since BUGSTONE performs intra-procedural analysis with a fixed security coding rule, each case is a single query-response exchange well within modern LLM context windows. We haven’t observed errors such as API retries or max-token errors, during the large-scale scan. The $110 figure represents the estimation of actual billed cost but not minimum. The costs in Table 5 are the actual cost of the experiments in Table 3.

---

### Official Review · Reviewer_2CD5 · 2026-03-10

**Soundness:** 2
**Presentation:** 2
**Significance:** 3
**Originality:** 3
**Overall Recommendation:** 5
**Confidence:** 4

**Summary:**

This paper introduces BugStone, an LLM-driven framework for Recurring Pattern Bugs (RPB) detection. The paper identifies the limitations of the LLM-based approach in the lack of accuracy and the static analysis approach in over-specification. Therefore, the paper proposes a hybrid approach to combine static and LLM-based analysis: it first uses static analysis to collect the code context of a patch, then uses an LLM to summarize the patch to pinpoint the bug root cause. For detection, it first utilizes a static analyzer to detect potential similar code implementations as candidates, then uses LLMs to analyze and identify whether the same issue fixed by the patch also occurs in the candidates. As a result, they have 246 validated bug issues and 34 confirmed and fixed.

**Compliance With Llm Reviewing Policy:**

Affirmed.

**Final Justification:**

The paper has a reasonable design and significant bug finding results (which impressed me the most), as mentioned in my initial review, making it quite interesting to me. The main concern I had was the missing details about the static analysis technique used, which the authors addressed in the rebuttal. Therefore, I decided to raise my final recommendation score to 5.

**Key Questions For Authors:**

- 1. Could you explain the details of static analysis implementation, as mentioned in W1?
- 2. Have you conducted any experimental comparison to quantify the improvement of your hybrid approach compared to pure static approaches, as mentioned in W2?
- 3. How did you use static analysis to detect logical bugs (not vulnerabilities like NPD) in your framework? For instance, if a bug is caused by failing to check the boundary of quicksort, how could you collect candidates by static analysis?

I’m fine to raise the score if all my concerns and questions are properly addressed.

**Limitations:**

- Function-level context
    - The paper only focuses on function-level contexts and excludes ~1000 patches due to multi-file dependencies. It’s an understandable limitation, but I’d hope the paper could discuss more about the trade-off between detecting multi-file triggered bugs and the context length issue.

I hope the paper can discuss more about their limitations, threats to validity, and improvements.

**Strengths And Weaknesses:**

## Strengths

### Significance
The paper makes a real-world impact by identifying more than 22K potential bugs in the Linux kernel and 47 command injection issues in Python projects, with 34 confirmed and fixed, and 246 validated. This is quite a valuable contribution to the community.

### Soundness
The paper’s observation of the limitations of static analysis and LLM detectors is reasonable, and it addresses these limitations by supplementing static analyzers with LLMs. This hybrid approach is overall sound.


### Originality
As mentioned, the insight of the paper about the limitations of static and LLM analyzers is valid, and how it designs the hybrid approach generally makes sense.


## Weaknesses

### Soundness
- **1. Missing details of static analysis.** The paper used static analysis for two purposes: 1. API function context extraction from the seed patch, and 2. candidate function extraction for violation detection. The paper didn’t mention details about these static analyses. First, I wonder how you use a LLVM-based static analyzer to extract function contexts. Are you using clang’s AST to extract the source code, or are you extracting LLVM IR? Second, the paper should include more details on how they use static analysis to identify candidate functions for later LLM-based violation detection. It is hard to evaluate the workflow comprehensively without any details provided.  Also, in the introduction,  the paper mentioned that “After that, equipped with a static program analyzer, BUGSTONE further detects potential similar code implementations along with their associated context.” Here, the “associate context” doesn’t seem to be explained in the methodology details.
- **2. Missing experimental comparison with pure static analysis.** The paper claims the limitations of static analysis, while lacking an ablation study on how the framework performs compared to pure static analysis, especially with a lack of details in the static analyzer used in its framework.

### Presentation
- The presentation is generally easy to follow. But please pay more attention to details. For instance, the paper marks the citations to green and table/figure/section references as red. However, only section numbers are marked red; for “(see Table 7)” at line 236 and “(Table 6)” at line 403, “Table” is not marked in red. Also, at around line 213, it mentions“Empirical results in Section §4 demonstrate that the Rule-Based approach yields the best trade-off between accuracy and cost”, while it seems the results are presented in Section 5.

---

> ### Author Rebuttal · Authors · 2026-03-30
>
> We appreciate your careful reading and constructive feedback. We respond to each question and concern below.
>
> ### For Q1:
> We apologize for the confusion caused by insufficient detail in the paper, which is mainly due to page limitations. We will incorporate these technical details upon acceptance.
> We employed two different approaches for API function context extraction and candidate function extraction respectively.
> For seed patch processing, we use "git diff W" to expand changed code into its full function context, and apply regular expressions to strip commit tags and code-semantics-irrelevant metadata. No LLVM or complex static analysis is involved, as compiling an arbitrary patch into LLVM IR is itself a non-trivial task and unnecessary, given that the goal is simply to provide the LLM with sufficient patch context. Further details are provided in Section 3.3 (Seed Patch Normalization).
> For candidate function extraction, we built an LLVM pass that operates on the kernel compiled to LLVM IR and then map to the source code positions. Specifically, we adapted and edited the state-of-the-art LLVM-based static analyzer MLTA (https://github.com/umnsec/mlta) to extract a complete call graph of the program. When a security coding rule is identified for a specific sink API function, the system uses this call graph to efficiently locate all call sites of that sink, extract the corresponding caller functions as candidates for scanning. For each candidate, the static analyzer retrieves its source-level position and extracts the corresponding source code with the LLVM IR metadata, which is then provided to the LLM for bug detection.
>
> ### For Q2:
> We want to answer this question from three perspectives.
> First, RPBs represent a new category of issues spanning multiple traditional bug types, such as null pointer dereference and memory leak. Traditional static analyzers are typically designed for a single specific bug pattern, each requiring substantial human effort to implement, maintain, and adapt to a specific target program. As a result, any individual static analyzer covers only a subset of the patterns, covered by BUGSTONE. A direct apple-to-apple comparison is therefore not straightforward.
>
> Second, we evaluate BUGSTONE against existing approaches using the ground truth dataset in Section 4.1.1, comprising thousands of bugs drawn from 23 top-tier security works, most of which rely on static or dynamic analyzers targeting the Linux kernel. The precision, recall, and accuracy results in Table 3 reflect BUGSTONE's performance on bugs found by these traditional approaches. Section 6.1 further demonstrates BUGSTONE's ability to detect previously unreported bugs representing cases missed by all prior approaches, including bugs newly introduced into the kernel after the publication of those studies.
>
> Third, we acknowledge that BUGSTONE exhibits a non-negligible false positive rate in large-scale detection. But it remains manageable for manual review and is comparable to rates reported by state-of-the-art static analyzers such as Crix[1].and APHP[2]. We also refer the reviewer to the concrete comparison with MoCQ in Answer 5 to the Reviewer PhbB for a detailed example.
>
> [1]. Lu, K., Pakki, A. and Wu, Q., 2019. Detecting Missing-Check bugs via semantic-and Context-Aware criticalness and constraints inferences. In 28th USENIX Security Symposium (USENIX Security 19)
> [2]. Lin, M., Chen, K. and Xiao, Y., 2023. Detecting API Post-Handling bugs using code and description in patches. In 32nd USENIX Security Symposium (USENIX Security 23)
>
>
> ### For Q3:
> We acknowledge that logic bugs without clear API or macro usage patterns, such as buffer overflows caused by incorrect index computation, fall outside the scope of the current work and represent a limitation of our approach. Our system is designed around recurring pattern bugs that are anchored to specific sink functions or API calls, which enables the static analyzer to construct a call graph and efficiently identify candidate functions. Bugs that lack such anchoring points, like the boundary check failure in the quicksort example, cannot be systematically surfaced through call graph traversal and are therefore not addressed in the current framework.
>
> ### Limitations of BUGSTONE:
> Beyond the limitations already discussed, we recognize that BUGSTONE currently operates primarily as an intra-procedural analysis tool, and bugs such as use-after-free that require inter-procedural reasoning or involve thread interactions are outside its current scope. A promising direction for future work is an agentic workflow that interleaves a static analysis engine for extracting relevant code slices with an LLM for semantic assessment. Additionally, BUGSTONE provides neither soundness nor completeness guarantees, as LLM outputs can be incomplete even for straightforward tasks. We expect these limitations to diminish as LLM capabilities continue to advance.

---

> > ### Author Rebuttal · Reviewer_2CD5 · 2026-03-31
> >
> > Thanks a lot for the authors' response, which addressed my concerns. I'll raise my score accordingly. Please make sure to clarify the details and scope more clearly when refining your manuscript.

---

> > > ### Author Response · Authors · 2026-04-06
> > >
> > > We thank you for the constructive feedback and review! We will address all unclear points and incorporate these suggestions to improve the clarity of the final manuscript.

---

### Official Review · Reviewer_VmFD · 2026-03-13

**Soundness:** 3
**Presentation:** 2
**Significance:** 3
**Originality:** 3
**Overall Recommendation:** 5
**Confidence:** 4

**Summary:**

The paper proposes an LLM-augmented system for the discovery of recurring pattern bugs. From one vulnerability sample, they use an LLM to abstract usage properties, find similar occurrences using static analysis and evaluate the findings with another LLM. Their evaluation is done on a manually constructed dataset and two real-world projects (Linux, top 100 Python repositories) showing the efficacy of their method.

**Compliance With Llm Reviewing Policy:**

Affirmed.

**Final Justification:**

The authors addressed my questions regarding the design of components in their rebuttal. Given the contributions of the paper and that these changes are editorial, I recommend accepting the paper.

**Key Questions For Authors:**

- What inputs does the static analyzer receive?
- What are the rules in Section 4.1.2 used for?

**Limitations:**

yes

**Strengths And Weaknesses:**

This paper presents a method for discovering recurring vulnerabilities in software repositories by leveraging the observation that a single root cause is often replicated across multiple locations within a codebase. The proposed approach operates in three stages: (1) analyzing the root cause of a vulnerability and its contextual characteristics, (2) performing static analysis to identify similar occurrences across the repository, and (3) verifying and prioritize the identified candidates using a large language model (LLM) to determine whether they are related to the original root cause.

To evaluate their approach, the authors construct a dataset of security patches and assess the system’s performance on this dataset. In addition, they apply the proposed method to the Linux kernel and the top 100 Python repositories, reporting discovered vulnerabilities, including confirmed instances.

**Contribution.** The paper addresses an important and practical problem in software security: identifying recurring vulnerabilities that arise from the same underlying root cause. The core observation that vulnerabilities frequently appear in multiple locations within a codebase is well motivated and supported by prior practice in vulnerability mining.

The proposed pipeline is conceptually appealing. In particular, combining traditional static analysis to identify candidate locations with an LLM-based verification stage is a promising design that may help reduce false positives, a common challenge in vulnerability detection systems.

The authors also evaluate the system on both a curated dataset of security patches and large real-world codebases such as the Linux kernel and widely used Python repositories. Demonstrating the applicability of the approach on real projects strengthens the practical relevance of the work.

Despite the interesting idea, several aspects of the method and evaluation remain unclear due to insufficient detail in the presentation.

**Methodological Details.** The description of the pipeline remains relatively high level and omits several important implementation details. For instance, the paper mentions the use of an LLVM pass for static analysis, but does not describe what this pass does or how it integrates with the rest of the system.

Similarly, the process by which information is fed into the static analyzer is not clearly specified. The paper refers to three components - root cause, required action, and scope constraints - but it is unclear how these components are represented and operationalized within the system.

Another missing detail concerns how the static analysis stage handles configuration-dependent code (e.g., preprocessor macros or #ifdef directives). Since such constructs are common in large systems like the Linux kernel, understanding how the analysis deals with them is important for evaluating the approach.

Providing more detail about the architecture and implementation of these components would significantly improve the clarity and reproducibility of the work.

**Evaluation Clarity.** The evaluation section is somewhat difficult to follow due to inconsistencies in how rule synthesis is described. In Section 3.3, security rules appear to be generated automatically by the LLM. However, Section 4.1.2 states that rules are extracted manually. It is unclear whether the manually extracted rules are only used for the HuRule evaluation or if they also play a role in the main experiments.

This ambiguity makes it difficult to understand exactly how the system is evaluated and how the different experimental settings relate to each other.

A clearer structure for the evaluation would help. In particular, separating the experiments into:
- a comparison against baselines, and
- an ablation study analyzing individual components of the system

would make the results easier to interpret. Currently, these aspects are intertwined and discussed as part of a single evaluation narrative.

This paper proposes a promising approach to identifying recurring vulnerabilities by combining static analysis with LLM-based verification. The core idea is interesting and the application to large real-world codebases demonstrates practical potential. However, the presentation lacks sufficient detail in several parts of the methodology and contains some inconsistencies in the evaluation description. Addressing these issues would significantly improve the clarity and reproducibility of the work.

Overall, I lean towards accepting the paper.

Minor Comments:
Page 6: The sentence “The formulas for these metrics are shown below.” appears to reference formulas that are not actually present in the text.

---

> ### Author Rebuttal · Authors · 2026-03-30
>
> We appreciate your careful reading and constructive feedback. We respond to each question and concern below.
>
> ### For Q1 regarding the inputs of the static analyzer:
> The static analyzer takes the entire target program as input, constructs a complete call graph, and records the source-level location and range of each function across the codebase. The primary purposes of the static analyzer are therefore to extract the call graph and to reduce the code scope that the LLM needs to examine.
>
> ### For Q2 regarding the rules in  Section 4.1.2:
> The 135 rules that mentioned in Section 4.1.2 are used for large-scale real-world bug detection, as further discussed in Section 6.1. These rules were derived from our ground truth dataset and known common issues in the Linux kernel. We then applied our system with these rules to detect previously unidentified bugs.
>
> ### Other Concerns: Confusion regarding rule synthesis and static analysis.
> We apologize for the confusion caused by insufficient detail in the paper.
> Regarding rule synthesis, human-written security rules are used as a baseline (HuRule rows in Table 3), serving only as one base line to quantify the performance differences between human-written and LLM-generated rules. In all other settings (by default), including the remaining rows in Table 3, such as Rule and Rule+Patch, and the real-world bug detection, BUGSTONE uses rules generated by the LLM from seed patches.
>
> The static analyzer is mainly used for candidate function and context extraction. We implemented an LLVM pass that operates on LLVM IR of the Linux kernel, leveraging the compilation process to naturally resolve macros, inline functions, and similar constructs before static analysis. It also retrieves the source-level line number ranges of every function via LLVM IR metadata.  Specifically, we adapted and edited the state-of-the-art MLTA analyzer (https://github.com/umnsec/mlta) to extract a complete call graph of the program. After that, when a security coding rule is identified for a specific sink function, BUGSTONE uses this call graph to locate all call sites of that sink, extract the corresponding caller functions’ source code as detection candidates. The extracted source code of each candidate is then provided to the LLM for bug detection. We will incorporate these technical details into the paper upon acceptance.

---

> > ### Author Rebuttal · Reviewer_VmFD · 2026-04-02
> >
> > Thank you for the response. My questions have been addressed.

---

> > > ### Author Response · Authors · 2026-04-06
> > >
> > > We thank you for the constructive feedback and review! We will address all unclear points as discussed in the final version.

---

### Official Review · Reviewer_7kzX · 2026-03-23

**Soundness:** 3
**Presentation:** 3
**Significance:** 3
**Originality:** 2
**Overall Recommendation:** 4
**Confidence:** 3

**Summary:**

BUGSTONE uses static analysis plus LLMs to turn one bug-fix patch into a generalized security rule, retrieve similar code sites, and detect recurring pattern bugs at scale. It introduces an 850-patch benchmark, reports 92.2% precision/79.1% pairwise accuracy, and finds many candidate issues in Linux and Python projects.

**Compliance With Llm Reviewing Policy:**

Affirmed.

**Key Questions For Authors:**

(1) Is the template in Figure 2 a rule template or a prompt template? Who provides the templates?

(2) Is the seed patch normalization done by LLM, or manually, or by means of certain algorithm?

**Limitations:**

Yes.

**Strengths And Weaknesses:**

Presentation:

The paper has clear definitions of important terms (Section 3.1), and gives concrete examples. Moreover, the paper uses the same example throughout the motivation and the methodology sections, which is very good.

There are still some points that can be improved.
(1) Is the template in Figure 2 a rule template or a prompt template? Who provides the templates?
(2) A simplified version of the rule table (Table 7) should be placed in the main text.
(3) Is the seed patch normalization done by LLM or manually or by means of certain algorithm?


Soundness:

The presentation is clear (the clear definition, the walk-through example, etc.).
Based on the clear presentation, the technique seems sound.

If the seed patch normalization is done manually, did the authors do the normalization for the 800+ patches? That seems to be a lot of human efforts.


Significance:

The problem addressed is significant and the proposed proposed approach seem quite practical.

The idea is straight-forward, but it works.


Originality:

The idea is straight-forward. The originality is not much.

---

> ### Author Rebuttal · Authors · 2026-03-30
>
> We appreciate the careful reading and feedback. We respond to each question below.
>
> ### For Q1:
> The "template" in Figure 2 refers to rule templates embedded in the prompt. We manually constructed three templates as few-shot examples to guide the LLM in generating security coding rules with the target function name, potential bug conditions, etc. (see Table 8). These templates serve to help the LLM better understand the preferred format of the expected security coding rules.
>
> ### For Q2:
> Patch normalization is performed by a deterministic algorithm. Specifically, we combine several Git commands, such as "git diff W", to extract the function-level context of each change and remove author information. We further apply regular expressions to strip commit tags and other metadata irrelevant to code semantics.

---

> > ### Author Rebuttal · Reviewer_7kzX · 2026-04-03
> >
> > Thank you for addressing my concerns. Now the rebuttal answers my questions. Please also put these clarifications/explanations in the paper so that it is much clearer.

---

> > > ### Author Response · Authors · 2026-04-06
> > >
> > > We thank you for the constructive feedback and review! We will incorporate these suggestions to improve the clarity of the final paper.

---

### Decision · Program_Chairs · 2026-04-30

**Decision:**

Accept (regular)

**Comment:**

This paper introduces BUGSTONE, a hybrid neuro-symbolic framework designed to detect Recurring Pattern Bugs (RPBs) at scale. The system leverages a single patched instance of a bug to synthesize natural language security coding rules via a Large Language Model (LLM). It then employs a static program analyzer (based on an LLVM pass) to identify candidate functions across a large codebase that match the context of the bug. Finally, an LLM verifies these candidates against the synthesized rules to confirm the presence of similar vulnerabilities. The authors demonstrate BUGSTONE's efficacy by identifying 246 valid bugs in the Linux kernel and discovering multiple vulnerabilities in top Python projects.


The reviewer consensus is generally positive (5, 5, 4, 3), with significant appreciation for the paper's practical impact. Reviewers were highly impressed by the large-scale deployment results, specifically the confirmation of hundreds of previously unknown bugs in the Linux kernel. The core idea of combining traditional static analysis (to prune the search space) with the semantic reasoning of LLMs (to verify bugs) was viewed as a technically sound and practical solution for a significant security problem. The use of a comprehensive dataset comprising thousands of bugs from 23 prior works to evaluate precision (92.2%) and pairwise accuracy provided a strong empirical foundation.


The primary concerns during the review period focused on implementation details of the static analyzer, the lack of head-to-head comparative baselines, and potential noise in real-world code. Several reviewers (VmFD, 2CD5) initially found the description of the LLVM pass and call-graph construction insufficient. The authors clarified their use of a modified MLTA analyzer to extract caller functions and source-level context, which satisfied these reviewers. Reviewer PhbB expressed persistent concerns regarding the absence of a direct comparison with tools like CodeQL or MoCQ on a shared codebase. The authors argued that such comparisons are difficult due to the specialized nature of traditional analyzers but pointed to their performance on the ground truth dataset (which includes bugs found by those tools) as a proxy. Concerns about LLM API costs were addressed with the clarification that a full Linux kernel scan (over 148k cases) cost approximately $110 USD, demonstrating the approach's economic viability.


While Reviewer PhbB remained "partially resolved" regarding the baseline comparisons and noise handling, the other three reviewers were satisfied by the authors' substantive technical responses and the clear evidence of real-world utility. The discovery of confirmed vulnerabilities in critical systems like the Linux kernel provides strong evidence of the framework's value. The authors have committed to incorporating the clarified implementation details and cost analysis into the final manuscript. Overall, this is a solid contribution to the field of automated bug discovery.